# Amorphous silicon-carbide photonics for ultrasound imaging
R. Tufan Erdogan [1] ✉, Bruno Lopez-Rodriguez [2], Wouter J. Westerveld [1], Sophinese Iskander-Rizk[1], Gerard J. Verbiest [1], Iman Esmaeil Zadeh [2] & Peter G. Steeneken [1] ✉

Photonic ultrasound sensors promise unparalleled spatial and temporal resolution in ultrasound imaging due to their size-independent noise figure, high sensitivity, and broad bandwidth. Optical materials can further improve performance and stability, but achieving small size, high sensitivity, and wide bandwidth remains challenging. This work introduces amorphous silicon carbide (a-SiC) for ultrasound sensing, offering strong optical confinement, low propagation loss, and high stability for miniaturized microring sensors. We demonstrate a compact detection system with a 20-transducers linear array coupled to a single bus waveguide. The sensors achieve an optical finesse of 1320 and intrinsic sensitivity of 78 fm kPa$^{-1}$, leading to a noise-equivalent pressure below 55 mPa/$\sqrt{\text{Hz}}$, calibrated from 3.36 MHz to 30 MHz. High-resolution imaging of fine structures validates real-world applicability. a-SiC is also easily integrated on most substrates due to its low deposition temperature. Our results position a-SiC as a promising solution for optical ultrasound sensing, combining miniaturization, low-loss, and high-sensitivity.

Medical ultrasound imaging is an indispensable tool for disease diagnosis. It enables capturing structural images of tissue deep under the skin non-invasively. For this purpose, pulses of ultrasound waves are sent deep into the tissue, and the echoes generated by reflections due to variations in acoustic impedance are recorded at the skin to create an image. However, because different tissues have almost the same acoustic impedance, these images contain little information on the variations in the type of tissue. Therefore, photoacoustic imaging (PAI) has been introduced, which gathers valuable information on the molecular composition of tissue by photo-selective generation of ultrasound waves, making it a promising candidate to become a mainstream imaging modality in clinical settings[1,2].

Reconstructing 2D or 3D acoustic or photoacoustic tomography (PAT) images requires recording the ultrasound wavefield at multiple locations[3]. This can be achieved either by scanning a single sensor across the area of interest or by using an array of sensors distributed over a plane. Current PAT imaging systems typically employ piezoelectric ultrasound sensor arrays or matrices to facilitate image reconstruction without the need for scanning[4]. However, piezoelectric detectors face inherent limitations in sensitivity, bandwidth, and scalability[5]. Their sensitivity decreases as their size is reduced, their bandwidth is limited by the piezoelectric response bandwidth, and each individual piezoelectric element requires a dedicated cable, which constrains integration density and application potential. To our knowledge, the smallest reported pitch for a piezoelectric detector array is

100 μm[6]. Since the spatial resolution in PAT is directly influenced by the detectors' element size, kerf, and bandwidth, which limits the spatial frequencies allowed in the grid, this restriction in piezoelectric sensors ultimately constrains the details in reconstructed images.

Although commercial piezoelectric hydrophones can detect high-frequency ultrasound signals and enable scanning ultrasound tomography with small scanning step sizes in a time-consuming fashion, they exhibit trade-offs in sensitivity when miniaturized. For example, commercial piezoelectric needle hydrophones with diameters of 0.04 mm and 0.2 mm have noise equivalent pressures (NEP) of 10 kPa and 1.1 kPa with operating bandwidths of 60 MHz and 40 MHz[7,8], respectively. In contrast, optical ultrasound detectors have demonstrated NEP values more than ten times lower[9,10] and operational bandwidths more than three times higher[9,11] compared to aforementioned hydrophones while maintaining sensor sizes smaller than 20 μm.

Accordingly, optical ultrasound detection using integrated photonic circuits is a promising technique with the potential to meet the PAI requirements in miniaturization, scalability, sensitivity, and bandwidth[12]. These sensors detect acoustic pressure via the effect on the refractive index and the deformation of the optical structure due to acoustic pressure[13]. To maximize detection performance while keeping the sensor size miniature, the optical response to acoustic pressure, such as resonance shift per impinging pressure, should be as high as possible. In addition, low optical

[1]Department of Precision and Microsystems Engineering, Faculty of Mechanical Engineering, Delft University of Technology, Delft, The Netherlands. [2]Department of Imaging Physics (ImPhys), Faculty of Applied Sciences, Delft University of Technology, Delft, The Netherlands. ✉e-mail: r.t.erdogan@tudelft.nl; p.g.steeneken@tudelft.nl

propagation loss and high finesse are critical parameters that directly influence the sensor's sensitivity, image resolution, and parallelization of the readout. Several optical waveguide geometries and materials were demonstrated using single miniature point-like sensors for integrated photonic ultrasound sensing. Polymer[14], silicon[15], and silicon-nitride[16] platforms have been employed to create an optical ring resonator and have proven the viability of integrated photonic ultrasound detection for photoacoustic imaging applications. Among these demonstrations, silicon waveguides with polymer top cladding are promising since their integration with the complementary metal-oxide-semiconductor (CMOS) fabricated electronics merely requires a small additional step of spin-coating. To this date, poly-dimethylsiloxane (PDMS), as a top cladding polymer, has been found to be a highly photoelastic material[17] under ultrasound pressure and is used in ultrasound sensing demonstrations[11,16,18]. Although single sensors could satisfy some of the needs in photoacoustic microscopy applications, a balanced demonstration of sensor arrays that address all requirements at once while maintaining small size, high sensitivity, and broad bandwidth, with a scalable, low-cost fabrication architecture, is still an open goal with high potential impact in photoacoustic tomography applications, hence the need for exploration of waveguide materials is required.

Recently, amorphous silicon carbide (a-SiC) has been introduced as an optical material for realizing integrated photonic ring resonators. It was demonstrated that a-SiC ring resonators possess high intrinsic-quality-factors, reaching $Q = 5.7 \times 10^5$ with propagation losses as low as 0.78 dB cm$^{-1}$ [19]. The a-SiC layer offers a refractive index $n = 2.55$ at 1550 nm wavelength, which can be adjusted by tuning its composition. The refractive index of a-SiC is higher than that of polymer, silicon nitride, and chalcogenide-based waveguides, but lower than that of silicon-on-insulator waveguides. The intermediate value of the refractive index of a-SiC offers a good compromise between size (higher $n$ enables smaller ring diameter) and propagation losses (smaller $n$ results in lower propagation losses).

Another advantage of a-SiC films is that they can be grown with high optical quality at temperatures as low as 150 °C. This allows a-SiC to be integrated in the backend-of-line of standard CMOS processes or on flexible material platforms that cannot withstand high temperatures[20,21]. This offers two prospects; firstly, integrating thousands of sensitive photonic ultrasound sensors with opto-electronic detection, acoustic signal processing, and imaging electronics in a single microchip can be achieved through CMOS integration. Secondly, the fabrication of the a-SiC waveguides can be performed on non-silicon substrates. For example, a-SiC waveguides can be grown on piezoelectric substrates[22] to simultaneously achieve photoacoustic signal detection and ultrasound pulse echo imaging; flexible substrates can be used to match the acoustic impedance of tissue or to integrate the backing layer in ultrasound applications. Additionally, transparent substrates, such as glass, can be used to allow photoacoustic excitation by light that passes through the sensor's substrate[23]. Furthermore, the nonlinear optical Kerr coefficient of a-SiC is ten times higher than that of silicon nitride and crystalline silicon carbide[24,25]. This nonlinear property allows generation of on-chip frequency combs[26], which could enable on-chip integrated parallel interrogation methods[27] and can increase the speed and imaging quality of these sensors while reducing their size.

Here, we introduce an optical ultrasound sensing application using the a-SiC waveguide platform. We demonstrate the potential of integrated optical sensors utilizing a-SiC ring resonators and resonator arrays by showcasing their performance in ultrasound tomography. For this purpose, we use a linear array of 20 a-SiC ring resonators, with a diameter of 16 μm at a pitch of 30 μm, that is coupled to a single bus waveguide. The a-SiC ring resonators feature a low noise equivalent pressure density (NEPD) below 55 mPa/$\sqrt{\text{Hz}}$ at a noise equivalent pressure (NEP) of 345 Pa, over a bandwidth of 26.6 MHz. Despite their small dimension (sensor radius of 8 μm), the ring resonators have a high intrinsic waveguide sensitivity (78 fm kPa$^{-1}$) and Q-factor ( $> 1 \times 10^5$). To demonstrate its potential for ultrasound tomography, we use the linear array to capture 2D cross-sectional ultrasound images of a 50 μm diameter aluminum wire and two 125 μm diameter glass

fibers, without scanning the position of the sensor array. Although the NEPD performance of the sensors can benefit improvements through laser noise canceling methods to achieve the state-of-the-art demonstrations, the achieved image quality provides confidence that the a-SiC platform is suitable for sensitive and broadband ultrasound imaging in advanced applications, such as photoacoustic tomography.

## Results

Figure 1a depicts the ultrasound characterization and imaging setup. The system enables ultrasound imaging with a-SiC circular microring resonator arrays, coupled to a straight bus waveguide. The light is coupled into and out of the chip using two focusing grating couplers, allowing access to the passive photonic circuitry through fibers positioned on both sides of the chip. The measurement setup includes a tunable laser to interrogate the sensors across different wavelengths. A photoreceiver connected to the output fiber converts the optical signals to electrical signals. The analog output from the photoreceiver is digitized using an oscilloscope triggered by the same arbitrary waveform generator that generates the transmitted ultrasound signals.

A microscope image of the polydimethylsiloxane cladded a-SiC ring resonator is provided in Fig. 1b. The resonator comprises a circular ring waveguide with a radius, $r$, and a straight bus waveguide placed at a gap distance from the circular ring resonator. The parameters defining the geometry of the single-ring resonator are provided in Fig. 1b, c. To evaluate and compare the optical performance and ultrasound sensitivity, we fabricated resonators with a waveguide width of 800 nm and a coupling gap ranging from 300 nm to 650 nm for ring radii of 6 μm and 8 μm at waveguide height 280 nm. Waveguide widths and heights of the resonator and the bus waveguides are kept the same throughout the study.

The main physical mechanism by which ultrasound can be sensed with integrated optical waveguides is the dependence of the refractive index of the waveguide materials on the incoming pressure. In the case of a highly photo-elastic polymer cladding, such as PDMS, the changes in the refractive index of the polymer cladding dominate the change in the overall effective index due to incoming pressure[11,18]. To assess this effective refractive index, in Fig. 1c, we provide a finite element method (FEM) simulation of the electric field of the TE0 mode and a schematic of the cross-section of the a-SiC waveguide used in this study. It comprises an a-SiC layer in the core, PDMS, and silicon dioxide in the top and bottom cladding layers, respectively. The simulation shows that the electric field distribution extends beyond the waveguide core boundary, retaining approximately 10% of its amplitude at a maximum distance of about  ± 500 nm from the center of the waveguide core for the plotted geometry. This evanescent field makes the waveguide sensitive to ultrasound pressure-induced changes in the refractive index of the PDMS via the photoelasticity of PDMS.

At the optical resonance wavelength, $\lambda_r$, of a ring resonator, its optical transmission $T$ shows a sharp resonance notch. To reach high-sensitivity levels, it is desirable to maximize the slope $|\frac{dT}{d\lambda}|$ at the flank of this notch. This optimization can be achieved for circular ring resonators by varying the coupling gap between the ring resonator and the straight waveguide. In Fig. 2a, b, the transmission response of ring resonators with different coupling gaps are plotted for ring radii of 6μm and 8μm, respectively. The slope of the ring resonance flank determines the optical sensitivity in ultrasound sensing. Sensors with small full-width-half-maximum (FWHM) and high extinction ratio (ER, see Eq. (3)) provide a higher slope on the flank of the ring transmission response. We plot each resonator's ratio ER/FWHM in Fig. 2c, as obtained from a Lorentzian fitting, and for each ring radius select the coupling gap that results in the highest ER/FWHM for ultrasound characterization experiments. There are discrepancies between the measured values and simulation results in Fig. 2c for some of the ring resonators. We attribute these variations to local differences in waveguide geometry and material optical properties that may result from the fabrication process including the possibilities of such as a variation of the optical properties in PDMS layer or a-SiC layer, entrapped particles/air bubbles in and around the ring to bus coupling region, electron beam lithography (EBL) patterning

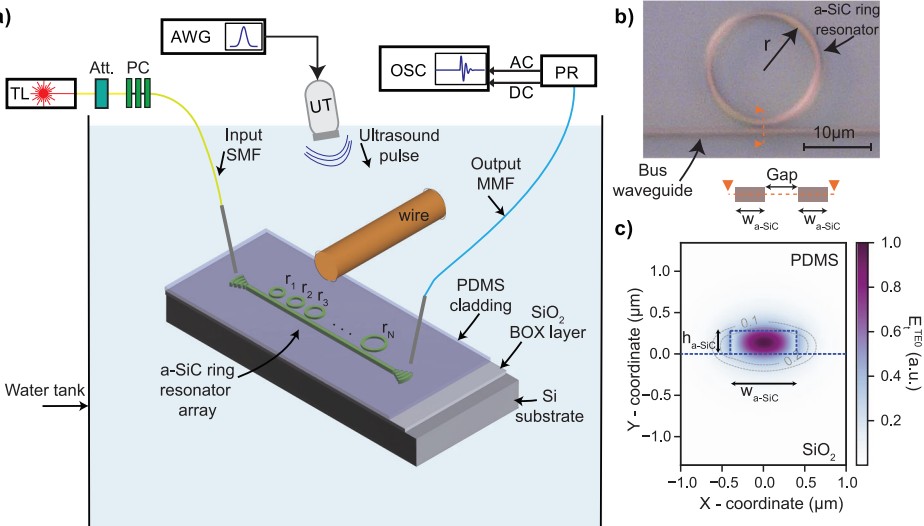

**Fig. 1 | Ultrasound sensing with an array of a-SiC ring resonators. a** Schematic of the integrated photonic sensor array chip and measurement setup. The setup uses the PDMS-clad a-SiC microring resonators in a water tank with grating couplers for optical interrogation. The sensed optical signals are converted into electrical outputs, which are then digitized in sync with the transmitted ultrasound. TL: tunable laser, Att: optical attenuator, PC: polarization control paddles, SMF: single-mode fiber, MMF: multi-mode fiber, AWG: arbitrary waveform generator, UT: ultrasound transducer, OSC: oscilloscope, PR: photoreceiver, AC, and DC: alternating and direct current, respectively. **b** Microscope image of a single PDMS-coated amorphous SiC micro ring resonator with $r = 8$ μm, $w_{\text{a-SiC}} = 0.8$ μm, and $gap = 0.45$ μm.

The cross-sectional schematic of the bus waveguide and ring waveguide is given at the bottom of the microscope image, and parameters associated with the cross-section are indicated. **c** Calculated total electric field amplitude of TE0-mode for the cross section with an a-SiC core height of 280 nm and a width of 800 nm. The material boundaries are indicated with blue dashed lines. The cover and bottom cladding materials are PDMS and $SiO_2$, respectively. The contour lines indicate the amplitude of the electric field, showing that the electric field is sensitive to variations in the cladding refractive index, primarily within approximately 500 nm of the waveguide core, with the sensitivity gradually diminishing beyond this region.

errors that may lead to excessive losses in the ring resonators at different locations of the fabricated chip. However, the overall trend of the optical performance of the resonators in Fig. 2c agrees with the simulation results. Further studies can investigate the local variations in the optical performance of the individual ring resonators. The error bars in Fig. 2c quantify the fitting errors up to two standard deviations.

Figure 2c shows that for rings with a radius of 6 μm, the ratio ER/FWHM is in general lower than that of the rings with 8 μm radius and that the optimal coupling gap occurs at smaller gap distances. We verify this observation with finite difference time domain simulations in Fig. 2c. To obtain the shown agreement between simulation results and experiments, the ring resonators' one round-trip amplitude losses are set to 3 dB cm$^{-1}$ and 9 dB cm$^{-1}$ for 8 μm and 6 μm ring radii, respectively. In Fig. 2d, the transmission spectrum measurement of the ring resonator with the highest ER/FWHM in our study is shown in detail. The resonator has a FWHM of 13.8 pm corresponding to a $Q$ factor of above $1.1 \times 10^5$ and an ER of 10.16 dB.

A high finesse in parallelized ring resonator arrays for ultrasound detection is important because it is directly related to the number of rings that can be simultaneously coupled and individually read out via a single bus waveguide. Fig. 2e provides a measurement of the free spectral range (FSR) of the ring with a radius of 8 μm. The distance between the two consecutive resonance modes in our sweep range is measured as 18.21 nm. According to this, the finesse of the resonator is calculated to be around 1300.

The schematic in Fig. 3a demonstrates the measurement principle of ultrasound sensing with optical microring resonators. The ring resonator has a resonance notch at the specific wavelength, $\lambda_r$. The impinging ultrasound signal changes the resonance wavelength of the microring resonator by $\Delta\lambda_r$. By interrogating the transmission of the ring resonator at a flank wavelength, $\lambda_f$, the ultrasound pressure-induced wavelength shifts can be converted to time-variations in optical transmission as is indicated by $\Delta T$ in Fig. 3a. The inset plots the transmission change over time due to the ultrasound signal received.

We select the ring resonators with the highest figure of merit (Fig. 2c) and characterize them under different ultrasound peak pressures. For these characterization experiments, the ultrasound transducer is excited with a Gaussian pulse at a central frequency of 19.6 MHz. The recorded signals as a response to this pulse from 8 μm radius ring for different calibrated peak pressures are plotted in Fig. 3b. The resonator's peak response increases as the pulse's peak pressure increases. Fig. 3c plots the absolute maximum value of the measured signal peaks given in Fig. 3b, demonstrating the linearity of the sensor's acoustic response at this pressure interval. The slope of the curve is found to be 0.24 μWkPa$^{-1}$. The pressure calibration procedure for the responsivity measurements is detailed in the Supplementary Note 4. The spectral responsivity, which is on average 9.7 mV kPa$^{-1}$ (see Supplementary Fig. 5c), is determined from the time domain signals by applying the Fourier transform. For this, we use the response of the sensor and a calibrated hydrophone's response to the same ultrasound signal with peak input pressure of 13.8 kPa. This spectral response is further used to evaluate the sensor's NEPD (see Supplementary Note 5).

To determine the NEPD of the selected sensors, as shown in Fig. 3d, we divide the noise amplitude spectral density of the resonator for each frequency by the spectral responsivity of the resonator. The noise amplitude spectral density is obtained from a time trace recorded without any form of averaging and without applying ultrasound pressure at the same flank wavelength and laser power as the measurements in Fig. 3b (see also[10] for the detailed methodology).

In Fig. 4a, a microscope image of an array of 20-ring resonators, coupled to a single straight bus waveguide, is shown. Each ring has a radius of approximately 8 μm and gap distance of 450 nm since these parameters gave the best NEPD performance in the single ring optical characterization experiments. The pitch (center to center distance) between the resonators is 30 μm, and each ring has a slightly different designed radius starting from 8.000 μm up to 8.116 μm, gradually increasing from the left of the image to the right with a step size of 5.8 nm. The waveguide width of the bus and ring waveguide is 800 nm.

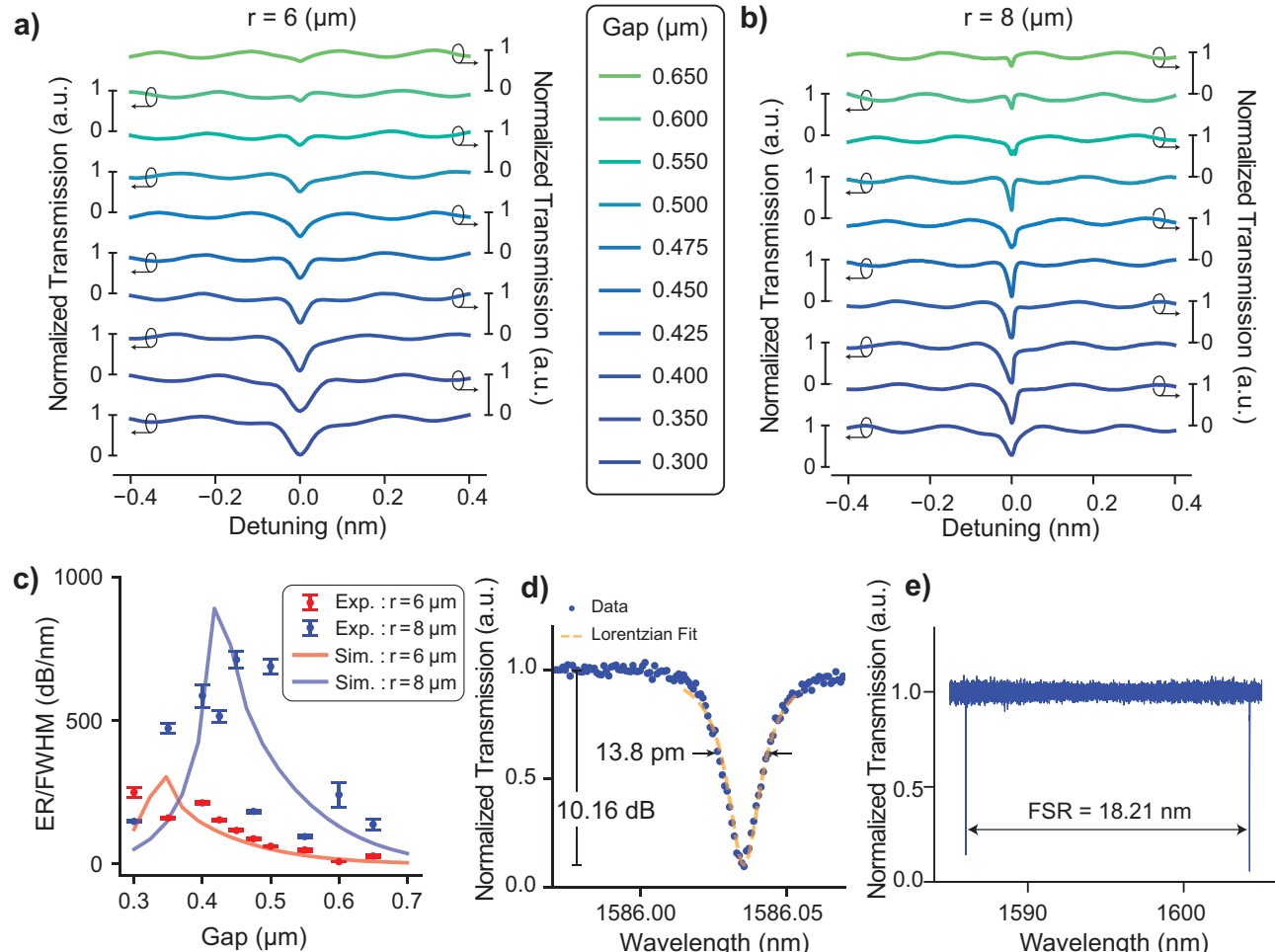

**Fig. 2 | Optical characterization of amorphous SiC single ring resonators.** The through-port transmission spectra of single-ring resonators for different coupling gaps with a radius of (**a**) 6 μm and (**b**) 8 μm, respectively. **a**, **b** share the same legend. **c** Comparison of the optical performance metric of the ring resonators with different coupling gaps and radii, together with the simulation results of the same geometries of the ring resonators. The ring with each radius' highest ER/FWHM ratio is selected for further ultrasound sensitivity analysis. Error bars correspond to Lorentzian fitting errors up to two standard deviations. **d** Transmission spectrum of the PDMS-coated a-SiC micro ring resonator with $r = 8$ μm, $w_{a\text{-}SiC} = 0.8$ μm, and $gap = 0.45$ μm. FWHM and ER measurements are annotated on the graph, and the values are extracted from the Lorentzian fit to the data. **e** FSR measurement of the single ring resonator in (**d**) from a wide range of wavelength sweep. The data is normalized to the background oscillations in the sweep measurement. The sweep range includes two consecutive resonance notches of the same resonator.

Figure 4b plots the laser wavelength sweep measurement for the array of 20 a-SiC ring resonators and for a single ring resonator. By design, the individual responses of all 20 rings are distributed within a single FSR. Each resonance notch within the FSR corresponds to a different ring resonator in the array, allowing us to read-out each ring passively, without requiring resonance wavelength tuning mechanisms such as heaters. Ultrasound measurements are performed using the 20 consecutive resonances as indicated in Fig. 4b. Due to random variations in fabrication, a sorting methodology is applied after the ultrasound measurements to match the positions of the rings in the chip with their resonance wavelengths in the spectrum sweep. The details of this sorting are provided in Supplementary Note 6.

Extracted resonance wavelengths of the ring resonators are plotted in Fig. 4c from the sweep measurement of the ring array against their determined positions from ultrasound measurements. We also repeated this sweep measurement 20 times for the linear array. In other words, we measured the resonance wavelengths of the array of 20 resonators 20 different times. The standard deviations in the measured resonance wavelengths from these 20 different sweep measurements of 20 different rings have an average of 13.84 pm. A maximum resonance wavelength deviation of 17.39 pm, and a minimum deviation of 12.2 pm for individual ring resonance wavelengths is observed within these measurements. It is also

important to determine the variation of the resonance wavelength distribution of the rings in the array from the intended linear distribution over one FSR. The standard deviation of the difference between measured resonance wavelengths from the linear fit is calculated to be 0.584 nm with a maximum difference of 1.45 nm. Most of the resonators align with a linearly increasing trend except the first four resonators, which may result from a local fabrication variability. Moreover, we observe significant non-uniformity of the ring notch responses in the linear array in terms of the FWHM and the ER of each ring resonator. This suggests that there is a significant variation in the coupling between the bus and the ring waveguide. Fig. 4d, e plot the results of FWHM and ER of each ring response in the array as a result of a Lorentzian fitting to their notch responses. We also determine the intrinsic sensitivity ($S_{int} = d\lambda_r/dP$) of the resonators (see Fig. 4f) using the methodology described in refs. 18,28. We analyze the factors that determine the intrinsic sensitivity in detail in Supplementary Note 2. We observe that the intrinsic sensitivity distribution over the array has a mean of 62 fm kPa$^{-1}$ and is between 19 and 95 fm kPa$^{-1}$. A theoretical calculation is provided in the Supplementary Note 2 and Supplementary Table 1, resulting in $S_{int} = 78$ fm kPa$^{-1}$, in good agreement with the measured values. The intrinsic sensitivity is governed by the pressure dependence of the resonance wavelength shift and not by the shape of the resonance notch. This difference in governing physics can explain the significant variations in values for

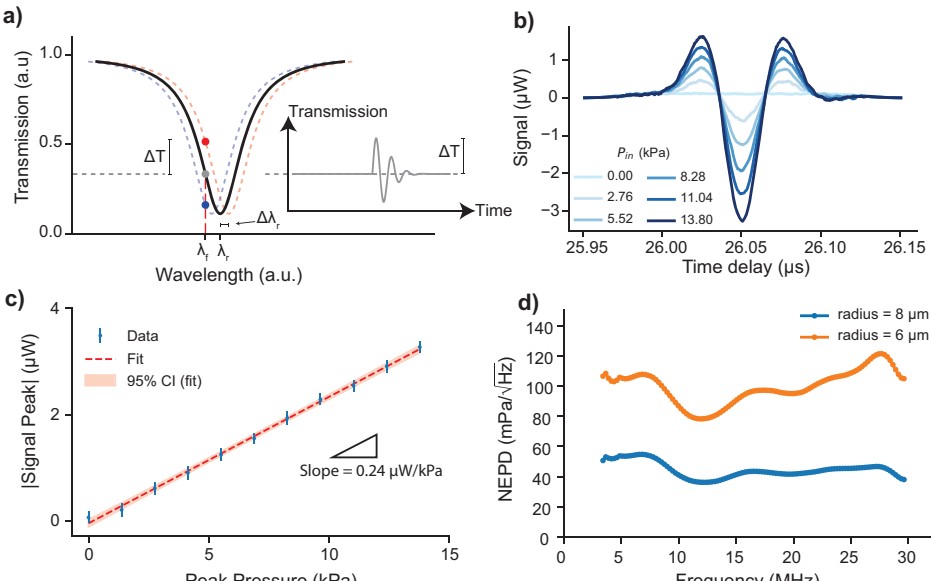

**Fig. 3 | Ultrasound sensitivity and NEPD characterization of amorphous SiC single ring resonators. a** Schematic of the read-out method by transmission output intensity monitoring when the laser wavelength is set to $\lambda_f$ on the flank of the unperturbed resonance spectrum (solid black line), the transmission amplitude is shown with a gray dot on the solid black line. The resonance wavelength without applied ultrasound pressure and wavelength shift for an applied pressure is indicated as $\lambda_r$ and $\Delta\lambda_r$, respectively. For a certain ultrasound pulse peak pressure, $\Delta T$ indicates the peak change in transmission of the ring resonance as a result of a $\Delta\lambda_r$ shift in resonance wavelength. The red and blue dashed lines correspond to the shifted resonance spectrum due to the applied ultrasound pulse, and the red and blue dots represent the transmission amplitudes when the wavelength is fixed at the flank wavelength. The inset schematically demonstrates the recorded sensor output transmission signal (solid gray line) in time when the laser is fixed at the $\lambda_f$ and an ultrasound pulse is received. **b** Recorded signals of the selected resonator under

ultrasound pulses with varying pressures when the laser wavelength is set to the flank wavelength of the resonator. The signals are averaged 100 times and cut from a longer trace using a Tukey window. **c** Plots the calibrated peak pressures against the absolute maximum of the measured signals, representing the linearity of the sensor in the range of applied pressures. The slope of the linear fit to the measured data is annotated in the plot area. The error bars correspond to the RMS value of the section of the recorded signal time trace preceding the ultrasound pulse. **d** NEPD of the sensor over the characterization bandwidth. The characterization bandwidth is limited by the bandwidth of the signal generated in the setup and received by the hydrophone. The lower frequency range limit is found by excluding the part of the full calibration spectrum where the signal amplitude reading from the hydrophone drops by 10 dB compared to its maximum. The high-frequency range limit is set to the maximum calibration frequency of the hydrophone used in calibration measurements (see Supplementary Note 4).

FHWM, ER, and $S_{int}$ in Fig. 4d–f, while the variations in $S_{int}$ seem to be relatively small.

To test the suitability of the array of a-SiC ring resonators for ultrasound tomography, we performed imaging experiments for two scenarios. Schematic cross-sectional views of the two experiments are given in Fig. 5a, b. For the first experiment, an aluminum wire with a measured diameter of 50.6 μm is placed in the water tank between the ring array and the ultrasound transducer, as shown in Fig. 5c. Two single-mode (SM) fibers with a diameter of 124.3 μm are placed at different heights for the second experiment, as shown on the micrograph in Fig. 5d. We use custom 3D-printed holders for these imaging experiments (see Supplementary Fig. 1). The ultrasound pulse generated by the ultrasound transducer at $t = 0$ travels in the negative Y direction, is reflected from the chip, then travels in the positive Y direction, reflects from the sample (aluminium wire or SM fiber), and then reaches the a-SiC array on the silicon chip for the second time. By analyzing both the first and second ultrasound pulses with the a-SiC array of microring resonators, we create an ultrasound image of the sample. We use this pulse-echo method to demonstrate the imaging performance of the sensor under a representative imaging scenario for the targeted ultrasound imaging applications.

The recorded time traces for both imaging scenarios (single aluminum wire or two SM fibers) are plotted in Fig. 5e, f. The time response of each of the 20 rings is included at the corresponding X-coordinate of the ring. We observe the initial arrival of the pulse to the ring array and the reflection signals from the aluminum wire and the SM fibers following the pulse signal after a time delay. The time-of-flight measurements agree well with k-wave simulations presented in Supplementary Fig. 2c and Supplementary Fig. 3c.

The resulting tomography images captured by a-SiC array are provided in Fig. 5g, h for the aluminum wire and the two SM fibers, respectively. The details about the image reconstruction algorithm are explained in the Methods section. The yellow curves in Fig. 5g are extracted from the image data at the location of the samples along the lateral and axial directions. The FWHMs of the single wire with 50.6 μm diameter in Fig. 5g along the X and Y directions are measured by using a Gaussian fit to the image signal and are found to be 49 ± 3.7 μm and 49 ± 3.4 μm, respectively. Uncertainty in the measured values is determined from the SNR of the fitted image data itself. The same procedure is also applied to the image of the SM fibers, and the FWHM of the image signals are 96 ± 7.6 μm and 161 ± 35.2 μm in Y direction; and 89 ± 11.5 μm and 139 ± 19 μm in X direction. The measurements agree with the sample diameter for the single aluminum wire. Although the results for SM fibers are found to be close to the actual diameter of the fibers, we attributed the difference to the imaging distance, hence the numerical aperture of the array at this depth of imaging. The numerical aperture of the array in Fig. 5g is 0.6 at the depth of the aluminum wire. On the other hand, in Fig. 5h, the numerical aperture becomes 0.28 and 0.2 for the SM fibers at depths of 0.95 mm and 1.3 mm, respectively. Thanks to the high finesse of a-SiC ring resonator platform, the accuracy of the measurement of the diameter and the imaging quality can further be improved by increasing the number of rings in the array, so that the numerical aperture will be higher for the same depth values.

## Discussion
We can theoretically compare the NEPD performance of the a-SiC ultrasound sensors to the widely used piezoelectric elements for ultrasound detection in photoacoustic imaging using the theoretical NEPD estimation

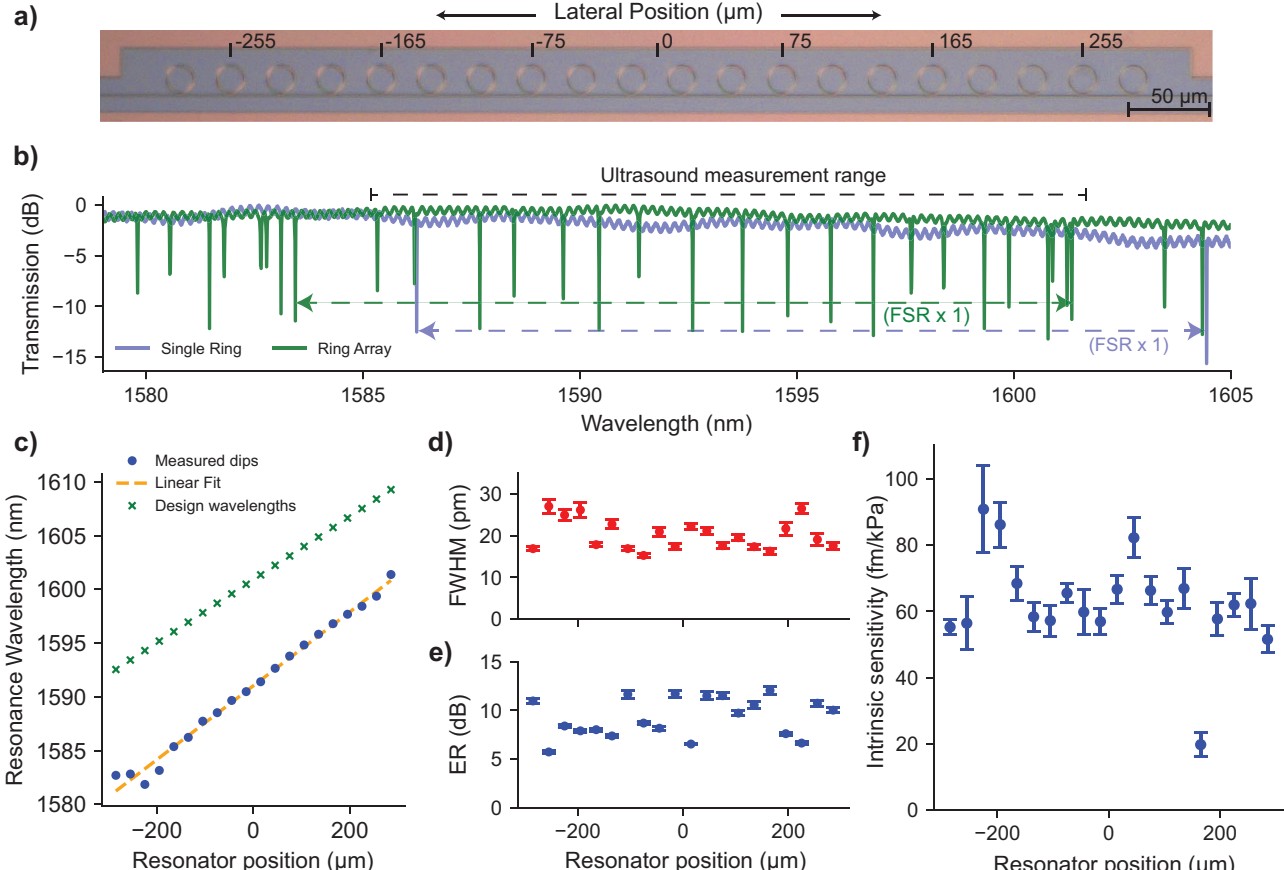

**Fig. 4 | Intrinsic sensitivity and optical characterization of an array of ring resonators coupled to single bus waveguide. a** Microscope image of the twenty rings used in ultrasound characterization experiments. The radius of each ring is slightly increasing from left to right of the image. The pitch of the array is 30 μm. The lateral axis is annotated, and a ruler is provided along the lateral axis over the array. **b** Measured spectra of the single ring resonator and the linear array of 20 a-SiC ring resonators. FSR and the range of ultrasound measurement notches are indicated with annotations. **c** The resonance wavelengths of the ring resonators are plotted against their lateral position. Resonator positions are determined using the ultrasound sensing and imaging experiments. A linear fit to measured resonance wavelengths is also plotted. Resonators follow a linear trend except for the first four resonators of the array. Intended design wavelengths of the resonators are also provided in the same plot. **d, e** Full-width half maximum (FWHM) and extinction ratio (ER) of each resonator in the array. Error bars in each sub-figure represent 2-sigma variation over the 20 different measurements of the same array with 20 resonators. **f** Intrinsic ultrasound sensitivity of each resonator in the array. This sensitivity is a measure independent of the resonator's linewidth and coupling effects. The error bars correspond to cumulative errors in responsivity measurement due to linear fitting, as well as errors in 20 different measurements of the slope of the resonator flank, up to a two-sigma deviation from the mean.

of piezo elements in photoacoustic imaging[29]. The NEPD of piezoelectric elements can be estimated using the equation:

$$\text{NEPD}(f)_{\text{piezo}} = \sqrt{\frac{F_n k T Z_a}{A \eta(f)}} \tag{1}$$

where $F_n$ is the noise factor of the preamplifier, $k$ is the Boltzmann constant, $T$ is the temperature, $Z_a$ is the acoustic impedance of the medium, $A$ is the area of the piezoelectric element, and $\eta(f)$ is the detector efficiency. Considering the conservative calibration range in our experiments, the high cut-off frequency is 30 MHz. For an ultrasound imaging array of transducers designed for this high-cut-off frequency, a half-wavelength matched piezoelectric transducer element should be of a size of 25 μm to avoid spatial aliasing. According to Eq. (1) and taking typical efficiency values for broadband transducers, $\eta(f)$, in the range between 0.01 and 0.001, the piezoelectric elements of this size theoretically have a NEPD between 45 mPa/$\sqrt{\text{Hz}}$ and 131 mPa/$\sqrt{\text{Hz}}$, respectively. On the other hand, a-SiC ultrasound sensor with a diameter of 16 μm performed with an average NEPD of 45 mPa/$\sqrt{\text{Hz}}$ on average in the range of our calibration (see Supplementary Note 5). Since we consider a theoretical piezoelectric element size matching the half wavelength of the high-cut-off frequency of

the ultrasound signal, there is no other metric to be introduced for this theoretical calculation. This is because there is no advantage for a piezoelectric transducer in being smaller than half of the smallest acoustic wavelength in the generated imaging signal[30].

The measured NEPD of our a-SiC platform is over 20 times better than that of the smallest hydrophones (1000 mPa/$\sqrt{\text{Hz}}$ at 40 μm diameter[7]), while occupying only one-fourth of their chip area. Furthermore, to our knowledge, the smallest demonstrated piezoelectric element size in a piezoelectric array of detectors is 100 μm, which makes them unsuitable for imaging at these higher frequencies or in near-field imaging scenarios without being affected by Nyquist sampling errors or compromising their sensitivity. Thus, a-SiC integrated photonic ultrasound sensors offer better sensitivity than piezoelectric elements without compromising the miniaturization in ultrasound sensing for high-frequency ultrasound signals.

To evaluate the array of a-SiC photoacoustic tomography sensors in comparison to the state-of-the-art of integrated photonic ultrasound sensor arrays, Table 1 lists the performance parameters of only the array of optical sensors designated for PAI applications.

Table 1 indicates that the demonstrated a-SiC resonator array showcases metrics that are competitive in all metrics while showing best-in-class performance in terms of finesse, pitch, and parallel detector count. It is of

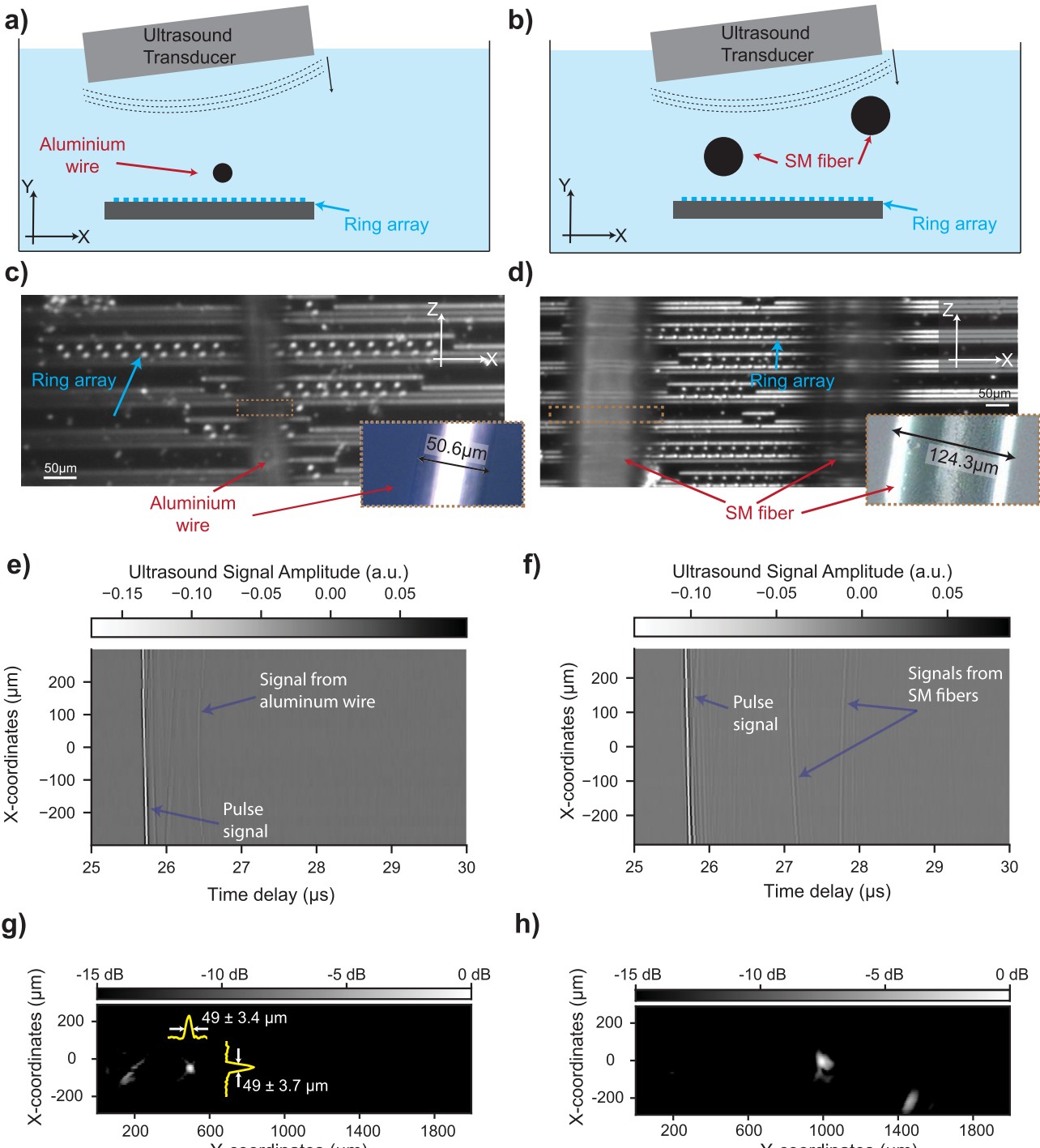

**Fig. 5 | Ultrasound tomography imaging demonstration with the array of a-SiC ring resonators. a, b** Schematic views of the ultrasound tomography experiments for 50 μm diameter aluminum wire and two of 125 μm diameter single mode fibers, respectively. The objects are in the water tank between the tilted ultrasound transducer and the array of rings at different heights. They are shown as a black circle, which corresponds to their cross-section. Inset plots the excitation signal of the ultrasound transducer to generate the ultrasound pressure wave. **c** Microscopy image of the ring array and the aluminum wire. The optical focus is adjusted to the chip plane. The inset shows the microscopy image of only the aluminum wire with higher magnification. **d** Microscopy image of the ring array and the two single-mode (SM) optical fibers. The optical focus is adjusted to the chip plane. The inset shows the microscopy image of one of the optical fibers with higher magnification. **e, f** Time traces of the signals in the a-SiC ring resonator array registered during the tomography experiments. The initial pulse arrives at different times at each sensor location and is correlated with the tilt angle. The signals from the aluminum wire and the SM fibers are indicated with arrows and text on the time traces. **g** Reconstructed image using frequency domain image reconstruction algorithm by using the time traces shown in (**e**). **h** Reconstructed image using frequency domain image reconstruction algorithm using the time traces shown in (**f**). The FWHM measurements of the wire in (**g**) and fibers in (**h**) are extracted from the yellow curves that are overlaid on the image by fitting a Gaussian function.

**Table 1 | Properties of the reported array of integrated photonic ultrasound sensors**

| Resonator | Platform | FWHM (pm) | Finesse | $S_{int}$ (fm kPa$^{-1}$) | Aperture | Pitch (µm) | Parallel detector count | Maximum film growth temperature (°C) | $\frac{S_{int}}{FWHM}$ (1/MPa) |
|---|---|---|---|---|---|---|---|---|---|
| $\pi$-BG[34] | SOI | 41.1 | N/A | 188 | 265 × 0.5 µm² | 130[a] | 5 | 1100 | 4.58 |
| Ring[10] | SOI-MEMS | 83 | **205** | **40000** | 20 µm | **30** | 10 | 1100 | **481** |
| Ring[36] | PS | 305 | – | 200 | 100 µm | 200 | 4 | **180** | 0.65 |
| Ring[28] | SOI | 130 | 147[d] | 40 | **10** µm | 250[b] | 8 | 1100 | 0.307 |
| Ring[27] | ChG | **2.6**[a] | – | **410**[c] | 40 µm | 400 | **15** | 350[37] | 157 |
| Ring | a-SiC | **13.8** | **1320** | 78 | **16** µm | 30 | **20** | **150** | 5.65 |

Best 2 performance metrics in bold face. Parameters that couldn't be found or calculated are left blank. SmartCut processing technology is assumed for SOI platforms for process temperature column[38].
*PS* polystyrene, *SOI* silicon on insulator, *N/A* Not applicable.
[a]Extracted from the published figure of the reference article.
[b]Includes integrated debugging and tuning interface blocks on the chip.
[c]Calculated from parameters in ref. 17 for a similar material.
[d]Calculated using reported parameters and the equation $Finesse = \frac{FSR}{FWHM}$ and $FSR = \lambda^2/n_g L$ where lambda is taken as 1550 nm, $n_g$ is the calculated group index of the TE0 mode for an SOI strip waveguide with FEM, and $L$ is the circumference length of the ring resonator.

interest to assess to what extent this performance is due to the material properties of the a-SiC core material. Therefore, we compare the theoretical performance of the a-SiC waveguides from this work to that of Si, silicon-nitride (SiN), and chalcogenide (ChG) waveguides used in ultrasound sensing from the literature in Supplementary Table 1. According to the Supplementary Table 1, the intrinsic sensitivity performance of our a-SiC platform is theoretically in between Si and SiN waveguides, thus providing a good balance between miniaturization and high sensitivity with $S_{int}$ twice higher than that of Si waveguides.

Another important aspect is the sensor's propagation and coupling losses, which determine its FWHM and finesse. From the low FWHM and high finesse in Table 1, the performance of the a-SiC waveguide platform boasts a very good loss performance compared to silicon and polystyrene devices. Several factors can contribute to this, which include high fabrication quality (layer roughness and waveguide uniformity), sufficiently large ring radius, optimized coupling gap (Fig. 2), and low optical absorption of the a-SiC material and cladding layers. The large free spectral range and low propagation loss of the a-SiC devices enable including many ring resonators on the same bus waveguide that can be read out individually or simultaneously[27].

Finally, the low growth temperature of the a-SiC of 150 °C is a clear advantage, especially when integrating them on platforms that cannot withstand high temperatures, like flexible substrates or the backend of CMOS wafers. Furthermore, a-SiC waveguides were demonstrated in heterogenous integration over lithium-niobate[22], which is a piezoelectric material that could enable generation of acoustic pulses on the same chip, or combined with SiN waveguides[31] to achieve extra functionalities such as on-chip frequency comb generation by making use of low-loss SiN waveguides and strong optical non-linearity to implement on-chip parallel interrogation for ultrasound sensing.

Nevertheless, the PDMS-cladded a-SiC platform in our work achieved a moderate NEPD performance compared to state-of-the-art integrated photonic micro-ring ultrasound sensors[10,27]. NEPD is not only dependent on the waveguide material/architecture but also affected by the optical interrogation instrumentation[32,33] and the interrogation methods employed[27,34]. In this study, to evaluate a-SiC platform, we employed the simplest interrogation method without further enhancements with noise cancellation strategies. Future studies on a-SiC platform may include differential optical interrogation techniques to enhance the NEPD of the measurements[27,33]. Moreover, using the combination of piezoelectric materials with a-SiC platform[22] holds strong potential to create advanced hybrid photonic-acoustic sensing platforms in future implementations.

## Conclusion

In this work, we introduce the first a-SiC photonic integrated ultrasound sensor array for imaging, featuring twenty ring resonators accessed through a single bus waveguide via wavelength-division multiplexing placed at a pitch of 30 µm. Our findings demonstrate a-SiC photonic circuits as a compelling platform for high-performance ultrasound sensing and acoustic tomography. With a high finesse of 1320 for a 16 µm diameter a-SiC ring resonator, we highlight the achievable optical quality along with the dense integration potential of this platform, opening the path toward ultra-dense 2D arrays with significantly enhanced imaging capabilities.

The calibrated maximum NEPD of 55 mPa/$\sqrt{\text{Hz}}$ across 3.36 MHz–30 MHz demonstrates competitive performance, while our analysis of intrinsic platform sensitivity of 78 fm kPa$^{-1}$ reveals that a-SiC nearly doubles the conventional silicon-core waveguide sensors' platform. This advantage provides a roadmap for further enhancing sensitivity by studying optimized cladding designs to increase intrinsic sensitivity, implementing noise cancellation methods for a-SiC sensors to decrease the NEPD of the sensors, and deploying a-SiC resonator arrays for photo-acoustic tomography applications.

Beyond its performance, a-SiC platform provides a unique combination of properties that are highly attractive for practical sensor deployment: narrow linewidths of resonators, miniature apertures, integration potential with piezoelectric substrate materials, and fabrication compatibility with CMOS electronics through low-temperature deposition (150 °C). These features not only enable miniaturized sensor arrays with exceptional spatial resolution and added functionalities, but also open opportunities for transformation into scalable, cost-effective ultrasound sensors for advanced imaging applications.

In conclusion, the results position a-SiC as a balanced material platform for the next generation of photoacoustic and ultrasound imaging. From high-resolution small animal brain imaging to intravascular catheter-based applications, a-SiC sensors offer the potential to deliver high performance in a miniature form, paving the way toward practical, high-density, and integrated optical ultrasound imaging systems.

## Methods

### Fabrication of amorphous silicon carbide ring resonators

Fabrication of the ultrasound sensors follows the same fabrication steps published earlier[19] at a low deposition temperature of 150 °C. For our application, the final silicon dioxide top cladding is substituted by an elasto-optic polymer, PDMS, with a measured thickness of around 5 µm. This is achieved by spin coating a PDMS mixing ratio of 1:30, and the detailed steps of the coating process are published earlier in ref. 18.

### Ultrasound characterization setup

We used the same setup published earlier[18] to perform the ultrasound characterization of the rings. In summary, we couple the laser light using two grating couplers; we submerge everything underwater as depicted in Fig. 1a.

We align the ultrasound source and launch Gaussian pulses directed to the sensor. By varying the peak-to-peak amplitude, we can characterize the sensitivity of the individual sensors and record the ultrasound wavefield by tuning the laser to the flank wavelength of each resonator in the array. The same setup is used for imaging experiments with an additional sample holder attached to the chip holder; sample holders for imaging are shown in Supplementary Fig. 1.

## Ultrasound image reconstruction

We use the sequentially recorded ultrasound data from each resonator in the linear array and the frequency domain reconstruction method to perform image reconstruction. In general, beamforming is performed by applying a 2D fast Fourier transform (FFT) to the received echoes, interpolating the data onto an acoustic dispersion grid, and then applying a 2D inverse FFT (IFFT) to reconstruct the signals. This method, described in ref. 35, enables Fourier-domain beamforming when using a plane wave for insonification. In more details, since the plane wave transmitted is angled, received data is first shifted in the frequency domain by multiplying by the propagator $e^{-i\omega\tau_{ch}}$ where $\tau_{ch}$ represents the individual channel delay steering the beam at the given angle $\Theta$, $\omega$ is the angular frequency, $\omega = 2\pi f$, here $f$ is taken as the center frequency of the transducer. Data is then interpolated in time to increase the sampling frequency and apodized by a 2D Tukey window as well as zero-padded (axially and laterally) to ensure smooth and sufficient frequency bins for further interpolation in the Fourier domain when applying a 2D FFT. Data is then remapped in the frequency domain into the object grid through complex shifting $k = \frac{k_y^2 + k_x^2}{2k_y \cos(\Theta) + 2k_x \sin(\Theta)}$, where $k_x = \frac{2\pi}{\text{pitch}}$ (with "pitch" being the element spacing), and $k_y = \frac{2\pi}{F_{\text{sampling}}}$. The beamformed image is then obtained after applying an IFFT.

## Optical characterization of ring resonators

To characterize the spectral response of a-SiC microring resonators, FWHM and ER of the single-ring resonators were extracted by fitting a Lorentzian function to the measured transmission spectrum. The measured transmission spectrum near the resonance was modeled and fitted using the Lorentzian function:

$$T(\lambda) = \frac{a_1}{(\lambda - \lambda_0)^2 + \left(\frac{\gamma}{2}\right)^2} + c \qquad (2)$$

where $\lambda$ is the wavelength, $a_1$ is the amplitude scaling parameter (it has a negative value), $\lambda_0$ is the resonance wavelength, $\gamma$ is the FWHM of the resonance dip, and $c$ is the baseline offset. Using the fit result, the ER was calculated as:

$$\text{ER}_{\text{dB}} = 10 \cdot \log_{10}\left(\frac{c}{\min(T)}\right) \qquad (3)$$

The loaded quality factor of the resonance was computed as: $Q = \frac{\lambda_0}{\text{FWHM}}$. This analysis was applied to all transmission spectra after low-pass filtering and normalization.

## Data availability

Data are available in the Supplementary Data file or from the corresponding author upon request.

## Code availability

Codes used in this study are available from the corresponding author upon request.

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

## Acknowledgements
I.E.Z. acknowledges funding from the European Union's Horizon Europe research and innovation program under grant agreement No. 101098717 (RESPITE project) and Dutch Research Council (NWO) OTP COMB-O project (18757). R.T.E. and P.G.S. acknowledge support from the PhoQuS-T project. The project (23FUN01 PhoQuS-T) has received funding from the European Partnership on Metrology (Funder ID: 10.13039/100019599), co-financed from the European Union's Horizon Europe Research and Innovation Programme and by the Participating States.

## Author contributions
R.T.E., G.J.V. and P.G.S. conceptualized the study and designed the experiments. B.L.R. and R.T.E. fabricated the devices. R.T.E. conducted the simulations, experiments, and data analysis/visualization. R.T.E. and S.I.R. implemented the ultrasound image reconstruction algorithm and visualization. R.T.E. and P.G.S. wrote the original draft. All authors contributed to the manuscript review and editing. W.J.W., S.I.R., G.J.V., I.E.Z., and P.G.S. provided the resources, funding and supervised the study.

## Competing interests
The authors declare no competing interests.
