## [Transparent Peer Review file · Communications Physics]

Amorphous Silicon-Carbide Photonics for Ultrasound Imaging

Corresponding Author: Mr Ramazan Erdogan

Version 0:

Reviewer comments:

Reviewer #1

(Remarks to the Author)

The authors have addressed my comments from their previous submission to Nature Communications. I would like to emphasize again that the paper is of high scientific quality and of interest to the research community. I still believe that for Nature Communications, the advantage of silicon carbide would need to be experimentally demonstrated (for example in a new application) and not just explained. Nonetheless, I agree that there is potential for advantages of using this material, as the authors explain. I therefore recommend the paper for publication in Communications Physics.

Reviewer #2

(Remarks to the Author)

After reviewing the revisions and the authors' responses to the points I raised in the previous round, I am satisfied that the issues have been adequately addressed.

Therefore, I recommend accepting the revised paper.

Response to reviewers' comments: Amorphous Silicon-Carbide

Photonics for Ultrasound Imaging

R. Tufan Erdogan^{1*}, Bruno Lopez-Rodriguez², Wouter J. Westerveld¹,
Sophinese Iskander-Rizk¹, Gerard J. Verbiest¹, Iman Esmail Zadeh²,
Peter G. Steeneken^{1*}

¹Department of Precision and Microsystems Engineering, TU Delft, The Netherlands.

²Department of Imaging Physics (ImPhys), TU Delft, The Netherlands.

*Corresponding author(s). E-mail(s): r.t.erdogan@tudelft.nl; p.g.steeneken@tudelft.nl;
Contributing authors: b.lopezrodriguez@tudelft.nl; w.j.westerveld@tudelft.nl;
s.iskander-rizk@tudelft.nl; g.j.verbiest@tudelft.nl; i.esmaeilzadeh@tudelft.nl;

1 Authors' response to reviewers' comments (Revision 1) :

In this section, Reviewer's comment is given in BLUE colored text, and the authors' response to the comment is given in BLACK colored text.

1.1 Reviewer 1:

In their manuscript, the authors demonstrate an ultrasound sensor array based on amorphous silicon carbide resonators coated with PDMS. I would like to acknowledge that the paper is well structured and that, indeed, very detailed meaningful research has been performed. Unfortunately, while the results are technically sound, I cannot recommend the paper for publication in Nature Communications for two main reasons: lack of innovation and limited performance.

We thank the reviewer for the kind words about our manuscript.

Lack of innovation: All the main ideas in the manuscript have already been published. This includes PDMS coating by Hazan et al and Pan et al and an array of micro-ring sensors sharing a single bus by Pan et al and Westerveld et al. The only new element in this work is the use of silicon carbide instead of silicon, silicon nitride, and chalcogenide glass. Indeed, silicon carbide can have some benefits like lower deposition temperature, which may allow creating flexible sensors, but it is the nature of things that some materials have advantages and disadvantages. There are numerous materials that one could use to reproduce the same concepts of resonator-based ultrasound sensing, all with their own mixtures of advantages and disadvantages. While it is important to learn what materials are compatible with these concepts, it is unreasonable to publish such findings in Nature Communications without something extra. To make their case, I think the authors should demonstrate a benefit from using a-SiC, e.g. actually producing a flexible sensor or on the surface of a piezoelectric transducer. Clearly, this would involve many more challenges than simply switching to a-SiC.

We thank the reviewer for their comments. We conclude from the feedback of this reviewer that, unfortunately, the innovation and challenges presented in our work were not adequately conveyed in our manuscript. There is certainly more innovation beyond just switching from conventional materials like Si or SiN to a-SiC. The reviewer is correct that 'it is the nature of things that some materials have advantages and others have disadvantages'. However, the number of materials that have sufficient advantages to compete with existing silicon and silicon nitride materials for integrated photonic ultrasound sensing is very small. Additionally, there are a few materials that could be CMOS-compatible (as a waveguide material) or can be added to CMOS fabricated dies through BEOL processes. Besides the potential of a-SiC waveguides to be integrated with CMOS fabricated electronics, the flexibility of the platform was recently demonstrated through integration on a piezoelectric

material surface [1]. Therefore, showing for the first time that a-SiC can provide comparable/competitive performance in integrated photonic ultrasound sensing is, in our view, a big innovation. To make the innovation and challenges presented in our work clearer to the reader, we have modified the manuscript as follows:

- Added sensor diameter and array pitch of the sensor array to the abstract.
- Page 3 Line 88: added the integration possibility of a-SiC waveguides with piezoelectric materials.
- Page 5 Line 96: Updated the paragraph to clarify the novelty of the manuscript.
- Page 14 Line 301: Updated the discussion section to include a demonstration of the flexibility of the platform.
- Page 15 Line 308: Added a paragraph to the discussion section about the NEPD performance in integrated optical ultrasound sensing, suggesting further improvements required for a-SiC sensors.
- Expanded the conclusion section to state the main findings of our work and achieved results.

Limited performance: while the authors demonstrate resonators with high Q-factors, the NEP is worse by over an order of magnitude compared to previous high-impact works in the field (Hazan, Pan, Westerveld), but no explanation is given. Imaging is performed for an aluminum wire, whereas Hazan and Pan performed in vivo imaging. While Pan performed parallel interrogation, here the interrogation is serial. Also, the demonstrated imaging resolution and sensor BW are quite conventional. This makes the work merely descriptive: it shows the performance achieved by a-SiC, but without demonstrating any benefits.

Yes, we agree with the reviewer's comment that the measured NEPD values of our platform are significantly below those mentioned current state-of-the-art demonstrations. Furthermore, NEPD is an important aspect, but not the only factor, that enables the sensors to achieve practical deployability in real-world applications for photoacoustic tomography [2] or microscopy [3]. NEPD is also a parameter that is dependent on the setup instrumentation and the method of interrogation [4, 5]. To clarify and inform the reviewer, [6, 7] employ another interrogation method to reduce the laser noise level to measure the sensor NEPD, which necessitates an extra reference delay line. In our study, we conducted the experiments using the simplest interrogation method to assess the platform's performance. Although it is clever to use laser noise reduction schemes and decrease the noise in the measurement, the comparison of our NEPD is non-equivalent to these works due to the added reference optical line. [8] utilizes a membrane to achieve record intrinsic sensitivity levels (see Table 1 of the main text), which creates a trade-off in fabrication ease and limits the sensor's bandwidth due to the mechanical properties of the membrane at the same time. In our work, the architecture is easy to fabricate, does not involve a mechanical membrane, and the switching waveguide to a-SiC does not limit the achievable bandwidth in ultrasound detection [9]. [7] utilizes a chalcogenide ring array that is grown at high temperatures, which limits its integration possibility with a CMOS circuit, whereas the a-SiC layer in our study is fabricated at 150 °C, which would not degrade CMOS-fabricated transistors if a-SiC is added through the BEOL processes. Of course, parallel interrogation of the ring resonators in [7] is a great method that is also applicable to a-SiC

ring resonator array by updating our lab's electronic/optical instrumentation to operate at tens of GHz. Lastly, [6] employs a π -BG to achieve the miniaturization in one direction, which limits the possible 2D pitch of the array by the length of the Bragg mirrors in the other direction (see main text Table 1 - Aperture), whereas in our work, we demonstrate miniature ring resonators that are easily scalable to a 2D array of resonators at the 30 μm pitch in both directions.

We demonstrated an imaging example that provides sufficient confidence to assess the performance of the sensors. Future studies in a-SiC can demonstrate photoacoustic *in vivo* or *in vitro* imaging, a densely packed 2D array of resonators.

Similar to our current view, our aim in improving the NEPD performance will consider solutions that do not compromise other performance parameters. For example, further increasing the intrinsic sensitivity by employing a highly photoelastic cladding material, without degrading optical quality or miniaturization (such as another polymer or another low-temperature-processed material with a similar RI to PDMS) could enable such an improvement. This is an interesting and active area of research. To deliver these nuances to the reader in our manuscript, we added some remarks (mentioned in the response to the first question) in the introduction and discussion sections of our manuscript on NEPD and future studies.

1.2 Reviewer 2:

The manuscript reports on the use of amorphous silicon carbide (a-SiC) as a material platform for optical ultrasound sensing, highlighting its favorable optical confinement, low propagation loss, and material stability. The work demonstrates a microring-based linear array of 20 transducers coupled to a single bus waveguide, achieving a maximum optical finesse of 1320 and an intrinsic platform sensitivity of 78 fm / kPa. High-resolution ultrasound imaging of fine structures is also presented, emphasizing the potential of a-SiC for miniaturized, low-loss, and high-sensitivity photoacoustic imaging applications. While the study presents clear technical merit and novelty, several critical issues need to be addressed to strengthen the manuscript: 1. The novelty of a-SiC as an optical ultrasound sensing material should be explicitly clarified, including whether this represents the first reported application.

We thank the reviewer for the kind comments and thorough evaluation of our manuscript.

To address the reviewer's suggestion, we have revised the paragraph in the main text on page 4, line 96. It now reads:

"Here, we demonstrate, for the first time, an optical ultrasound sensing application using the a-SiC waveguide platform. We demonstrate the potential of integrated optical sensors utilizing a-SiC ring resonators and resonator arrays by showcasing their performance in ultrasound tomography. For this purpose, we use a linear array of 20 a-SiC ring resonators, with a diameter of 16 μm at a pitch of 30 μm , that is coupled to a single bus waveguide. The a-SiC ring resonators feature a low noise equivalent pressure density (NEPD) below

55 mPa/ $\sqrt{\text{Hz}}$ at a noise equivalent pressure (NEP) of 345 Pa, over a bandwidth of 26.6 MHz. Despite their small dimension (sensor radius of 8 μm), the ring resonators have a high intrinsic waveguide sensitivity (78 fm kPa $^{-1}$) and Q-factor ($> 1 \times 10^5$). To demonstrate its potential for ultrasound tomography, we use the linear array to capture 2D cross-sectional ultrasound images of a 50 μm diameter aluminum wire and two 125 μm diameter glass fibers, without scanning the position of the sensor array. Although the NEPD performance of the sensors can benefit improvements through laser noise canceling methods to achieve the state-of-the-art demonstrations, the achieved image quality provides confidence that the a-SiC platform is suitable for sensitive and broadband ultrasound imaging in advanced applications, such as photoacoustic tomography.”

2. The frequency range applicability, particularly for lower-frequency ultrasound (e.g., tens of kHz), should be discussed to demonstrate the versatility of the material.

We thank the reviewer for the question. In our study, we did not experimentally explore the limit frequencies well below (below 3 MHz) or far above (above 30 MHz) the central frequency of our ultrasound transducer ($f_c = 19.6$ MHz). The purpose of this is to provide a valid pressure calibration through the use of the certified calibration of the hydrophone and a single ultrasound transducer. Thus, the characterization of the operation bandwidth is limited by the ultrasound transducer and the calibration of the hydrophone.

3. Discrepancies between simulation and experimental results (Fig. 2) should be explained, accounting for fabrication tolerances, material property variations, or experimental uncertainties.

We thank the reviewer for the question.

In Fig. 2c of the main article, the comparison of measured values to simulated values illustrates the trend of the effect of coupling strength via varying gap, and explains our strategy for finding a resonator with a high ER/FWHM ratio to enhance the sensor performance. The reviewer is correct that discrepancies exist between the simulations and the measured ER/FWHM values for different gaps. However, all the ring resonators with different coupling gaps were placed as one copy in the fabricated chip, so any random variation in the ring resonator response due to fabrication, material nonuniformity, either in PDMS or a-SiC, will affect the measured FWHM and ER. This is why we fabricated several rings with different gap values for each ring radius, allowing us to understand the overall trend for optimization and ultimately select the resonator with the best figure of merit for further ultrasound characterization. Even so, the overall trend somewhat matches between the simulation and the measured samples, which suggests to us a range of gap values that would provide the best responsivity in ultrasound sensing for a given waveguide geometry and ring radius for the a-SiC platform.

We added an explanatory paragraph, as shown below, to the main text on page 6, line 142, to suggest future studies on optimizing fabrication variations. It now reads:

”There are discrepancies between the measured values and simulation results in Fig. 2c for some of the ring resonators. We attribute these variations to local differences in waveguide geometry and material optical properties that may result from the fabrication process including the possibilities of such as a variation of the

optical properties in PDMS layer or a-SiC layer, entrapped particles/air bubbles in and around the ring to bus coupling region, EBL patterning errors that may lead to excessive losses in the ring resonators at different locations of the fabricated chip. However, the overall trend of the optical performance of the resonators in Fig. 2c agrees with the simulation results. Further studies can investigate the local variations in the optical performance of the individual ring resonators.”

To provide further insight to the reviewer about these variations, we calculated the ring radius distribution using the measured distribution of resonance wavelengths. The resonance wavelengths of the rings in the main text Fig. 2c have a standard deviation of 0.427 nm and 0.625 nm for rings with a radius of 6 μm and 8 μm , respectively. This corresponds to a standard deviation in radius of 1.6 nm and 3.14 nm for rings with radius of 6 μm and 8 μm , respectively. This variation in radius can be explained by the tolerance in the EBL patterning process. However, we believe that uniformity of the fabrication process is not in the scope of this work. Future studies could explore and optimize the uniformity of each ring on the fabricated chip.

Furthermore, the variation of the FWHM and ER in the linear array of rings can also be observed in the main text, Fig. 4b, 4d, and 4e. Although they are located within 600 μm distance, their FWHM varies from 14 pm to 29 pm, and their extinction ratio varies from 5.5 dB to 13 dB, whilst these rings have the same waveguide width and coupling gap, with a slight difference in ring diameter. This variation was already discussed in a paragraph on page 11, line 212 of the main text.

4. The noise-equivalent pressure density (NEPD) and associated noise level need to be clarified, either by providing measured noise data or detailing the calculation procedure.

The calculation procedure of NEPD was provided in Section 5 of the SI text, and we added reference articles to the same SI section. The NEPD calculation of the sensors follows the earlier work in the field [7, 8]. The measured noise data for the ring resonator and the calculated noise amplitude spectral density from these time series noise measurements for the ring with a radius of 8 μm were provided in Fig. S5d and Fig. S5e, respectively.

5. Sensor long-term stability is not addressed and should be discussed or tested, given its importance for practical applications.

Fig. R1 Optical characterization comparison for long-term stability investigation. The measurement dates are printed at the top of the plots.

To verify the long-term stability of the sensors, we repeated optical and ultrasound measurements using the same ring resonator with a diameter of 16 μm , which was also used in our manuscript. In Fig. R1, we plot the wavelength sweep results of the same ring resonator, which was maintained in our lab for approximately 13 months within a petri dish; the lab is equipped with an air conditioning system, temperature, and humidity control. Further tests could be performed to evaluate performance under thermal or other environmental stress factors; however, we believe that these extensive reliability tests are not within the scope of the current work. We observe that the resonance wavelength has shifted by 326 pm, FSR has decreased by 20 pm, FWHM of the fitted Lorentzian varied by 0.08 pm, and ER is varied by 1.31 dB. These results suggest a slight change in the optical properties of the materials over a period of more than one year and within the fitting and coupling errors from measurement to measurement.

Fig. R2 Long-term in ultrasound sensing performance. **a)** and **b)** The recorded ultrasound signals using 16 μm diameter ring resonator for different pressures. **c)** and **d)** extracted peak response amplitudes and fit results.

Fig. R2 and Fig. R3 demonstrate the long-term ultrasound sensing stability of the a-SiC ring resonator. We used the RMS value of the baseline noise traces in Fig. R2a and Fig. R2b as the standard deviation of the signal peaks that are plotted in Fig. R2c and Fig. R2d, respectively. The confidence intervals of the linear fit slope measurements for different pressures are also given as shaded regions in Fig. R2c and Fig. R2d. We observe that the RMS noise level in the measurements on the left column of Fig. R2 is higher than the RMS noise of the measurements in Fig. R2d. This is due to the DC bias difference in the recorded signals. Additionally, the responsivity to ultrasound is slightly different for the two measurements; however, we attribute this to the slight difference in optical parameters between the two measurements.

In Fig. R3a and Fig. R3b, we provide the noise ASD of the long-term performance comparison experiments and in Fig. R3c and Fig. R3d, the resulting NEPD spectrums for the two measurements are plotted. We observe that maximum and mean NEPD levels remain similar with no significant difference observed after more than a year.

As a result of these comparison experiments in both optical and ultrasound domains, we conclude that the PDMS-cladded a-SiC ultrasound sensor continues to deliver similar performance after a year.

Fig. R3 Long-term change in ultrasound performance

6. Figures 3(c) and 4(c) should include error bars to indicate measurement uncertainty and reproducibility.

We thank the reviewer for the kind suggestions. We calculated the RMS value of the baseline noise in the recorded ultrasound signals preceding the pulse peak for each recorded signal at each pressure, as shown in Fig. R3a. This baseline RMS value is calculated and plotted as the error bars in the new version of the figure for each signal recorded for each pressure. Additionally, we added the confidence interval of the linear fit. In this letter, the initial version and the current version of the main text Fig. 3c are given side by side in Fig. R4.

In Fig. 4c of the main article, the resonance wavelengths for each resonator are found by fitting a Lorentzian to the wavelength sweep measurement of the linear array. The error in the resonance wavelength determination is small compared to the scale of the y-axis in the main text Fig. 4c. If plotted as error bars for each resonator, the error bars become invisible in the plot. Hence, we exclude the error bars in the figure. To report the variation in these measurements, we modified the paragraph in the main text page 11 line 202 as below:

Fig. R4 Updated sensor response figure in main text Fig. 3c with error bars and fit confidence. **LEFT** Initially submitted version. **RIGHT** Updated current version with error bars and fit confidence interval.

”Extracted resonance wavelengths of the ring resonators are plotted in Fig. 4c from the sweep measurement of the ring array against their determined positions from ultrasound measurements. We also repeated this sweep measurement 20 times for the linear array. In other words, we measured the resonance wavelengths of the array of 20 resonators 20 different times. The standard deviations in the measured resonance wavelengths from these 20 different sweep measurements of 20 different rings have an average of 13.84 pm. A maximum resonance wavelength deviation of 17.39 pm, and a minimum deviation of 12.2 pm for individual ring resonance wavelengths is observed within these measurements. It is also important to determine the variation of the resonance wavelength distribution of the rings in the array from the intended linear distribution over one FSR. The standard deviation of the difference between measured resonance wavelengths from the linear fit is calculated to be 0.584 nm with a maximum difference of 1.45 nm. Most of the resonators align with a linearly increasing trend except the first four resonators, which may result from a local fabrication variability.”

7. The conclusion section should more explicitly highlight the most significant results, including any transformative improvements in sensitivity, detection limit, or response time.

We thank the reviewer for addressing this. We have extended the conclusions section in our manuscript, highlighting the most significant results based on the findings of our study.

In summary, the manuscript presents promising results and demonstrates significant potential, but addressing the above points is essential for establishing confidence in the reported performance I therefore recommend that the manuscript be reconsidered after a major revision.

We thank both reviewers for their time and feedback.

2 Authors' response to reviewers' comments (Revision 2) :

Reviewer 1 (Remarks to the Author):

The authors have addressed my comments from their previous submission to Nature Communications. I would like to emphasize again that the paper is of high scientific quality and of interest to the research community. I still believe that for Nature Communications, the advantage of silicon carbide would need to be experimentally demonstrated (for example in a new application) and not just explained. Nonetheless, I agree that there is potential for advantages of using this material, as the authors explain. I therefore recommend the paper for publication in Communications Physics.

Reviewer 2 (Remarks to the Author):

After reviewing the revisions and the authors' responses to the points I raised in the previous round, I am satisfied that the issues have been adequately addressed. Therefore, I recommend accepting the revised paper.

We thank both reviewers for their time and feedback that led us to revise and improve our manuscript.

References

- [1] Li, Z. *et al.* Heterogeneous integration of amorphous silicon carbide on thin film lithium niobate. *APL Photonics* **10**, 016120 (2025). URL <https://doi.org/10.1063/5.0228408>.
- [2] Garrett, D. C. & Wang, L. V. Acoustic sensing with light. *Nature Photonics* **15**, 324–326 (2021). URL <https://www.nature.com/articles/s41566-021-00804-z>.
- [3] Prebeck, A. *et al.* Comparison of bulk piezoelectric and opto-mechanical micromachined detectors for optoacoustic and ultrasound sensing. *IEEE Sensors Journal* **25**, 34459–34467 (2025).
- [4] Riobó, L. *et al.* Noise reduction in resonator-based ultrasound sensors by using a cw laser and phase detection. *Opt. Lett.* **44**, 2677–2680 (2019). URL <https://opg.optica.org/ol/abstract.cfm?URI=ol-44-11-2677>.
- [5] Lee, Y. *et al.* Theoretical and experimental study on the detection limit of the micro-ring resonator based ultrasound point detectors. *Photoacoustics* **34**, 100574 (2023). URL <https://linkinghub.elsevier.com/retrieve/pii/S2213597923001271>.
- [6] Hazan, Y., Levi, A., Nagli, M. & Rosenthal, A. Silicon-photonics acoustic detector for optoacoustic micro-tomography. *Nature Communications* **13**, 1488 (2022). URL <https://www.nature.com/articles/s41467-022-29179-7>.
- [7] Pan, J. *et al.* Parallel interrogation of the chalcogenide-based micro-ring sensor array for photoacoustic tomography. *Nature Communications* **14**, 3250 (2023). URL <https://www.nature.com/articles/>

s41467-023-39075-3.

- [8] Westerveld, W. J. *et al.* Sensitive, small, broadband and scalable optomechanical ultrasound sensor in silicon photonics. *Nature Photonics* **15**, 341–345 (2021). URL <https://www.nature.com/articles/s41566-021-00776-0>.
- [9] Bao, K. *et al.* Photoacoustic imaging sensors based on integrated photonics: Challenges and trends. *Laser & Photonics Reviews* **19**, 2400414 (2024). URL <https://onlinelibrary.wiley.com/doi/abs/10.1002/lpor.202400414>.